# Correlation functions of three-dimensional Yang-Mills theory from the FRG

Lukas Corell[1], Anton K. Cyrol[1], Mario Mitter[2], Jan M. Pawlowski[1,3] and Nils Strodthoff[4]

**1** Institut für Theoretische Physik, Universität Heidelberg,
Philosophenweg 16, 69120 Heidelberg, Germany
**2** Department of Physics, Brookhaven National Laboratory, Upton, NY 11973, United States
**3** ExtreMe Matter Institute EMMI, GSI, Planckstr. 1, D-64291 Darmstadt, Germany
**4** Nuclear Science Division, Lawrence Berkeley National Laboratory, Berkeley, CA 94720, USA

## Abstract

We compute correlation functions of three-dimensional Landau-gauge Yang-Mills theory with the Functional Renormalisation Group. Starting from the classical action as only input, we calculate the non-perturbative ghost and gluon propagators as well as the momentum-dependent ghost-gluon, three-gluon, and four-gluon vertices in a comprehensive truncation scheme. Compared to the physical case of four spacetime dimensions, we need more sophisticated truncations due to significant contributions from nonclassical tensor structures. In particular, we apply a special technique to compute the tadpole diagrams of the propagator equations, which captures also all perturbative two-loop effects, and compare our correlators with lattice and Dyson-Schwinger results.



# 1   Introduction

Functional methods such as the Functional Renormalisation Group (FRG) or Dyson-Schwinger equations (DSEs) are non-perturbative first-principles approaches to Quantum Chromodynamics (QCD), and they are complementary to lattice simulations. At finite density the latter approach is hampered by a sign problem, while the former approaches face convergence and accuracy problems. The aim of the fQCD collaboration [1] is to establish the FRG as a quantitative continuum approach to QCD, with the phase diagram and the hadron spectrum as primary applications, see [2–7] for recent works.

Building on the advances made in a previous work in four-dimensional space-time [5], we consider Landau-gauge YM theory in three dimensions, in this work. Similar to its four-dimensional analogue, it is asymptotically free and confining. Upon adding an adjoint scalar, it corresponds to the dimensionally reduced asymptotic high-temperature limit of four-dimensional YM theory. Furthermore, the reduced dimensionality allows lattice simulations at a considerably reduced numerical expense, making the three-dimensional theory an interesting testing case that allows truncation checks in functional approaches. Therefore, the propagators of three-dimensional YM theory have been studied intensively on the lattice [8–24], with DSEs [25–31], and in semi-perturbative settings [32–34]. Its vertices have been investigated on the lattice [11,13] as well as with continuum methods [29,31,34].

So far, the most advanced results for YM theory in three dimensions within functional approaches have been obtained in a recent DSE investigation [31]. There, the coupled system of equations for the classical tensor structures has been solved self-consistently. In terms of the complexity of the truncation, the investigation [31] is comparable to the calculation performed in [5] for the four-dimensional case, which is more complicated due to non-trivial renormalisation. The present work builds on these advances, with a focus on the effects of including non-classical vertices and tensor structures in the tadpole diagrams of the gluon and ghost propagator equations.

The paper is organized as follows: In Sec. 2 we review the treatment of YM theory with the FRG using a vertex expansion for the effective action. We focus on new developments for the inclusion of the propagator tadpole diagrams. In Sec. 3 we discuss our results, which includes a thorough investigation of apparent convergence and a comparison to DSE and lattice results. The conclusion is given in Sec. 4. We check the independence of the regulator and describe the computational setup in the appendices.

# 2   Yang-Mills Theory from the FRG

In this section we review the FRG approach to YM theory using a vertex expansion for the effective action. Although the overall set-up follows [5,35], we repeat the most important steps for the convenience of the reader.

The FRG is a non-perturbative continuum method that implements Wilson's idea of including quantum fluctuations in momentum shells for the effective action, see [36–40] for QCD-

$$\frac{\partial \Gamma_k}{\partial t} = \frac{1}{2} \, \bigotimes - \bigotimes$$

Figure 1: Flow equation. Wiggly and dotted lines represent the dressed gluon and ghost propagators, respectively. The crossed circles denote regulator insertions $\partial_t R$, see (1).

related reviews. The key object in this approach, pioneered by Wetterich [41], is the scale-dependent analogue of the effective action $\Gamma_k$. The RG or infrared cutoff scale $k$ is introduced via a momentum-dependent regulator function $R_k$ that acts like a fluctuation-suppressing mass term on momentum scales $p^2 \lesssim k^2$. The scale dependence of $\Gamma_k$ is governed by an exact equation with a simple one-loop structure,

$$\partial_t \Gamma_k[\phi] = \frac{1}{2} \int_p G^{ab}_{\mu\nu}[\phi] \, \partial_t R^{ba}_{\nu\mu} - \int_p G^{ab}[\phi] \, \partial_t R^{ba} \,, \tag{1}$$

where $\int_p = \int \mathrm{d}^3 p/(2\pi)^3$ and the full field-, momentum-, and scale-dependent gluon and ghost propagator

$$G_k[\phi] = \frac{1}{\Gamma^{(2)}[\phi] + R_k} \,, \qquad \Gamma^{(n)}_k[\phi] = \frac{\delta^n \Gamma[\phi]}{\delta \phi^n} \,. \tag{2}$$

The superfield $\phi = (A_\mu, c, \bar{c})$ consists of gauge, ghost, and anti-ghost fields. In (1) the propagators $G^{\mu\nu}_{ab}[\phi]$ and $G^{ab}[\phi]$ are the diagonal gluon and off-diagonal ghost–anti-ghost components of the propagator (2). A pictorial representation of (1) is given in Fig. 1.

The regulator functions are given in App. A, where we also demonstrate the independence of the results from the choice of the regulator function. Flow equations for the 1PI $n$-point functions are straightforwardly derived from (1) by taking functional derivatives with respect to the fields, see Fig. 2 for the diagrammatic equations.

## 2.1 Vertex expansion

Due to the structure of the flow equation (1), the flow equation for an $n$-point correlator depends on up to $(n+2)$−point functions. This leads to an infinite tower of coupled equations, which have to be truncated within appropriate non-perturbative expansion schemes in order to be numerically solvable. As in [5], we work in a systematic vertex expansion scheme, corresponding to an expansion of the effective action in terms of 1PI correlation functions. Relying on the structural similarities of the three-dimensional theory to its four-dimensional analogue, we take all classical vertices into account, i.e. the ghost-gluon, three- and four-gluon vertex. In addition, we compute so-called tadpole vertices as discussed in the next subsection 2.2. For later reference we quickly recapitulate the parametrisations for the propagators and classical vertex functions considered in this work. The gluon and ghost two-point functions are parametrised in terms of scalar dressing functions $1/Z_A(p)$ and $1/Z_c(p)$,

$$[\Gamma^{(2)}_{AA}]^{ab}_{\mu\nu}(p) = Z_A(p) p^2 \delta^{ab} \Pi^\perp_{\mu\nu}(p) \,,$$

$$[\Gamma^{(2)}_{\bar{c}c}]^{ab}(p) = Z_c(p) p^2 \delta^{ab} \,, \tag{3}$$

where $\Pi^{\perp}_{\mu\nu}(p) = \delta_{\mu\nu} - p_{\mu}p_{\nu}/p^2$ denotes the transverse projection operator. We parametrise the three-point vertices by

$$[\Gamma^{(3)}_{\bar{c}cA}]^{abc}_{\mu}(p,q) = \sqrt{4\pi\,\alpha(\mu)}\,\lambda_{\bar{c}cA}(p,q)\,[\mathscr{T}^{\text{cl}}_{\bar{c}cA}]^{abc}_{\mu}(p,q)\,,$$

$$[\Gamma^{(3)}_{A^3}]^{abc}_{\mu\nu\rho}(p,q) = \sqrt{4\pi\,\alpha(\mu)}\,\lambda_{A^3}(p,q)\,[\mathscr{T}^{\text{cl}}_{A^3}]^{abc}_{\mu\nu\rho}(p,q)\,. \tag{4}$$

Their classical tensor structures are given by

$$\left[\mathscr{T}^{\text{cl}}_{\bar{c}cA}\right]^{abc}_{\mu}(p,q) = \mathrm{i}f^{abc}q_{\mu}\,,$$

$$\left[\mathscr{T}^{\text{cl}}_{A^3}\right]^{abc}_{\mu\nu\rho}(p,q) = \mathrm{i}f^{abc}\left\{(p-q)_{\rho}\delta_{\mu\nu} + \text{ perm.}\right\}\,. \tag{5}$$

The transversely projected basis for the ghost-gluon vertex consists of only one single element, whereas the corresponding basis for the three-gluon vertex counts four elements. The impact of non-classical tensor structures in the three-gluon vertex have been found to be subleading [42] in four space-time dimensions. Here we assume that they are also subleading in three dimensions and neglect them. The parametrisation of the four-gluon vertex is given by

$$[\Gamma^{(4)}_{A^4}]^{abcd}_{\mu\nu\rho\sigma}(p,q,r) = 4\pi\,\alpha(\mu)\,\lambda_{A^4}(\bar{p})\,[\mathscr{T}^{\text{cl}}_{A^4}]^{abcd}_{\mu\nu\rho\sigma}\,, \tag{6}$$

where the classical tensor structure is given by

$$\left[\mathscr{T}^{\text{cl}}_{A^4}\right]^{abcd}_{\mu\nu\rho\sigma} = f^{abn}f^{cdn}\delta_{\mu\rho}\delta_{\nu\sigma} + \text{perm..} \tag{7}$$

The inclusion of non-classical tensor structures in the four-gluon vertex is discussed below in subsection 2.2. The four-gluon dressing function(s) are approximated as a function of the average momentum $\bar{p}^2 = \frac{1}{4}(p_1^2 + p_2^2 + p_3^2 + p_4^2)$ which was shown to be a good approximation for the full momentum dependence in four space-time dimensions [43] and we assume that the same holds in three dimensions.

From the momentum-dependent dressing functions of the different correlators, we can define corresponding running couplings via

$$\alpha_{\bar{c}cA}(p) = \alpha(\mu)\frac{\lambda^2_{\bar{c}cA}(p)}{Z_A(p)\,Z_c^2(p)}\,,$$

$$\alpha_{A^3}(p) = \alpha(\mu)\frac{\lambda^2_{A^3}(p)}{Z_A^3(p)}\,,$$

$$\alpha_{A^4}(p) = \alpha(\mu)\frac{\lambda_{A^4}(p)}{Z_A^2(p)}\,. \tag{8}$$

Due to gauge invariance, encoded in the Slavnov-Taylor identities, all the couplings (8) have to agree in the perturbative regime of the theory. Furthermore, the dimensional suppression of the running coupling ensures that the dressing functions take their bare values at large momentum scales,

$$\lim_{p\to\infty}\lambda_{\bar{c}cA}(p) = \lim_{p\to\infty}\lambda_{A^3}(p) = \lim_{p\to\infty}\lambda_{A^4}(p) = 1\,, \tag{9}$$

for UV-trivial wave function renormalisations

$$\lim_{p\to\infty}Z_A(p) \to 1\,, \qquad \lim_{p\to\infty}Z_c(p) \to 1\,. \tag{10}$$

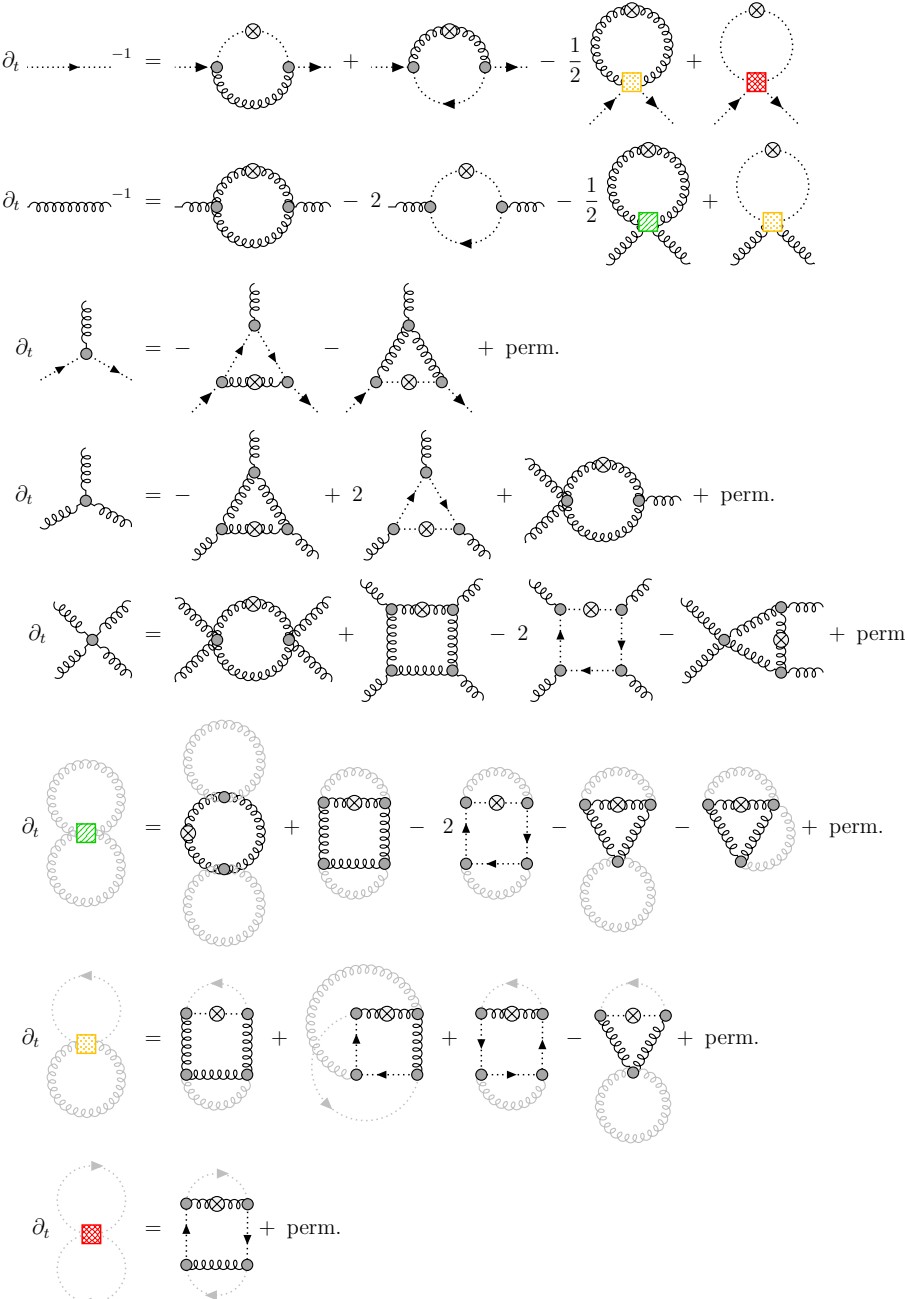

Figure 2: Diagrams that contribute to the truncated flows of propagators and vertices. While filled circles denote dressed (1PI) vertices, the squares denote the tadpole vertices explained in subsection 2.2. Shaded lines indicate the projection procedure of the tadpoles vertices. Permutations include not only (anti-)symmetric permutations of external legs but also permutations of the regulator insertions.

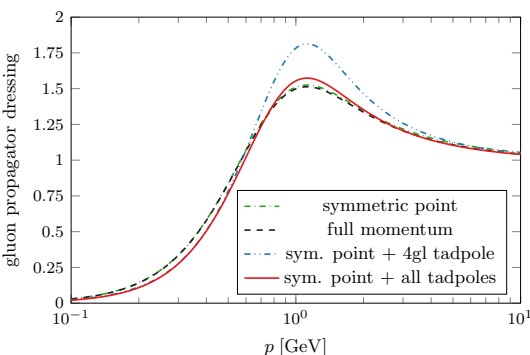
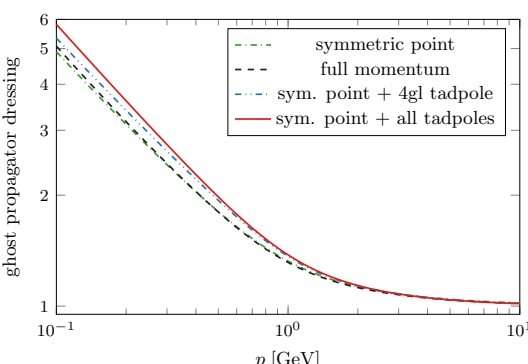

Figure 3: Truncation dependence of the gluon propagator dressing $1/Z_A(p)$ (left) and ghost propagator dressing $1/Z_c(p)$ (right). *Symmetric point* and *full momentum* denotes using the average momentum and full momentum dependency, respectively, in the three-gluon vertex. Results with *+ 4gl tadpole* and *+ all tadpoles* include the respective tadpole diagrams.

The truncation described above depends only trivially on the gauge group. In particular, only the quadratic casimir of the adjoint representation appears in the truncated set of equations. Therefore, it can be absorbed into a redefinition of the coupling, which in turn can be turned into a redefinition of the physical scale, see [7,35] for a more detailed discussion. The same holds for the extended truncation described in the next subsection. Thus, our results are effectively independent of the gauge group. However, this does not indicate a bad truncation since also in perturbation theory YM theory is independent of the gauge group up to three loops, see e.g. [44] for a recent discussion. Also the DSE results from [31] do not possess a genuine gauge group dependence and lattice results for the propagators show only a mild dependence on the gauge group [45,46]. Consequently, we compare our results to $SU(2)$ lattice results.

## 2.2 Tadpole vertices

The structure of the flow equation (1) implies that fully dressed four-point functions appear on the right hand side of the propagator equations, see Fig. 2. In general, this requires the full knowledge of all momentum-dependent non-classical four-point tensor dressings. Although some exploratory studies exist [43,47–50], their dynamical back-coupling into the propagator equations has still not been achieved. In the following, we propose a method that captures most of the dynamics on the level of the propagator equations, while it keeps the numerical effort at a manageable level. As an example, we consider the gluon tadpole contribution to the gluon propagator equation. All other tadpole diagrams are obtained analogously. The gluon tadpole contribution to the flow of the gluon two-point function is given by

$$\partial_t [\Gamma^{(2)}_{A^2}]^{ab}_{\mu\nu}(p) = \frac{1}{2} \int_p [\Gamma^{(4)}_{A^4}]^{abcd}_{\mu\nu\rho\sigma}(p,-p,q) \cdot [G \, \partial_t R \, G]^{dc}_{\sigma\rho}(q) . \tag{11}$$

Exploiting that the gluon propagator is diagonal in colour space and transverse with respect to its momentum in Landau gauge, we can project (11) with $\delta^{ab} \Pi^{\perp}_{\mu\nu}(p)$. From this we see that the gluon propagator equation depends only on the projected four-point function

$$T_{A^4}(p,q) = \Pi^{\perp}_{\mu\nu}(p)[\Gamma^{(4)}_{A^4}]^{abcd}_{\mu\nu\rho\sigma}(p,-p,q) \Pi^{\perp}_{\rho\sigma}(q) . \tag{12}$$

Therefore, the full contribution of the four-gluon vertex to the tadpole is already contained in this single scalar function, whose flow we can compute directly from projecting the corresponding equation accordingly, cf. Fig. 2. In particular, this procedure includes the back-coupling

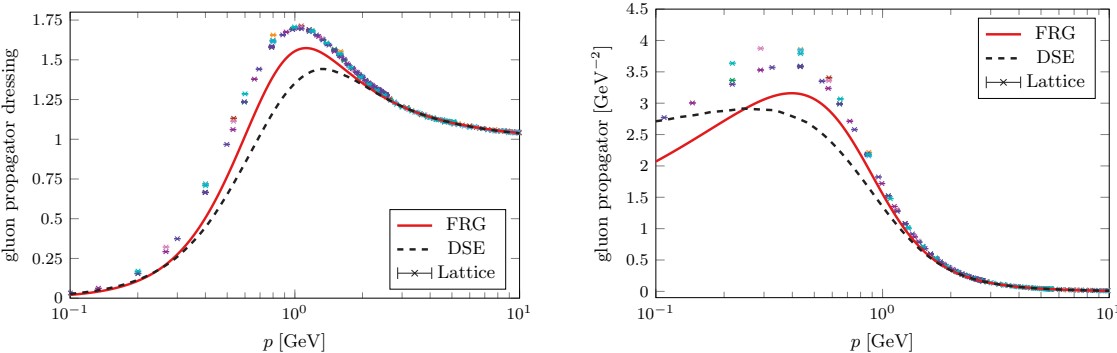

Figure 4: Gluon propagator dressing $1/Z_A(p)$ (left) and the dimensionful propagator $1/(p^2 Z_A(p))$ (right) in comparison with DSE [31] and lattice [11, 13, 23, 51] results.

effect of all non-classical tensor structures that are generated at the perturbative one-loop level, including therefore also all two-loop effects of the tadpole diagrams in the propagator equations. The non-classical tensor structures couple back into the vertices indirectly via the propagators. We neglect their direct back-coupling into the vertex equations. However, we expect this approximate treatment to yield a considerable improvement of the truncation at comparably moderate numerical costs.

## 3 Results

In this section we present the main findings of our investigation. Our solutions are of the scaling type, and are obtained as described in App. B. After discussing the truncation dependence of our results we provide an extensive comparison to results from lattice gauge theory and Dyson-Schwinger equations. We close with a determination of the infrared scaling coefficients and their comparison to those of finite temperature Yang-Mills theory in four dimensions.

### 3.1 Truncation and Apparent Convergence

In order to assess the influence of the truncation on our results, we compare three different extensions of our simplest symmetric point approximation:

1. *symmetric point*: only classical vertices with dressing functions that depend only on the symmetric momentum configuration,

2. *full momentum*: same as 1. symmetric point, but including the full momentum dependence of the ghost-gluon and three-gluon vertex dressings,

3. *sym. point + 4gl tadp.*: same as 1., but with the effects of the non-classical tensors of the four-gluon-vertex included in the tadpole diagram of the gluon propagator equation as described in subsection 2.2,

4. *sym. point + all tadp.*: same as 3., but additionally including the effects of the two-ghost-two-gluon and four-ghost vertices in both propagator equations, see subsection 2.2 and Fig. 2 for a visualization.

The corresponding results for the propagators are shown in Fig. 3. The first immediate observation is that the additional momentum dependence (2.) in the three-gluon and ghost-gluon

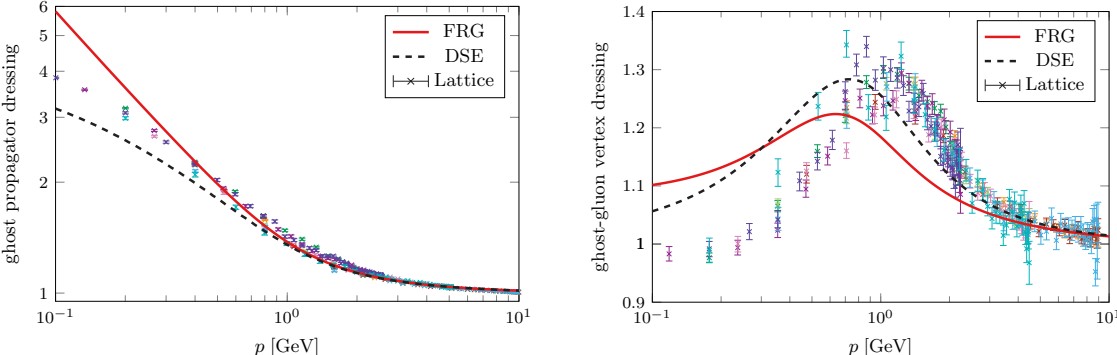

Figure 5: Ghost propagator dressing $1/Z_c(p)$ (left) and ghost-gluon vertex dressing $\lambda_{\bar{c}cA}(\bar{p})$ (right) compared to DSE [31] and lattice [11,13,23,51] results.

vertices does not visibly affect the propagators. On the contrary, the full momentum dependence and tensor structures of the four-point functions in the tadpole diagrams significantly affect the propagators. Concerning the goal of apparent convergence, we observe that including the tadpole contribution of the four-gluon vertex alone has a comparably pronounced effect, most of which is counteracted by the remaining tadpoles. This indicates that a fast convergence may be achieved if the underlying consistent resummation pattern is preserved within the truncation scheme. A similar observation has already been made in the matter sector of QCD in four space-time dimensions [2,6]. There, it is found that the effect of non-classical tensor structures in the quark-gluon vertex is counter-acted by corresponding structures in higher quark-gluon interactions that stem from the same BRST-invariant operator. We conclude that it is of chief importance to fully reveal these resummation patterns.

## 3.2 Comparison to DSE and Lattice

In this section we compare the results from our most extensive truncation, 4. *sym. point + all tadp.* (see subsection 3.1), to results obtained from $SU(2)$ lattice gauge theory [11, 13, 23, 51] and with Dyson-Schwinger equations [31]. To that end we normalise both, lattice and DSE results respective to our results in the UV regime, for more details see App. C. We emphasise again hat the presented FRG result is of the scaling type [52–60], whereas the lattice and DSE results are decouplings solutions [12,14,17,61,62], characterised by a finite, non-vanishing value of the gluon propagator at $p = 0$.

### 3.2.1 Propagators

From Fig. 4 and the left panel of Fig. 5, it is clearly seen that our results agree well with the rescaled lattice results in the UV regime with a discrepancy arising below 3 GeV. This difference is most likely due to truncation artifacts in our results which has to be clarified in future work. The most obvious culprit are missing effects in the equations for the classical vertex tensor structures due to the leading non-classical tensor structures of the three- and four-point functions.

The DSE gluon propagator from [31] has a smaller bump than both the FRG and lattice propagators. In subsection 3.1 we have shown that non-classical tensor structures have the net effect of increasing the bump in the gluon propagator. In comparison to the DSE truncation in [31], the present approximation includes more non-classical tensor structures. Although this may serve as an explanation, the system of equations is highly non-linear, and such an incomplete comparison is potentially misleading. Another factor may be that the DSE results are of the decoupling type whereas our results are of the scaling type, which generically show

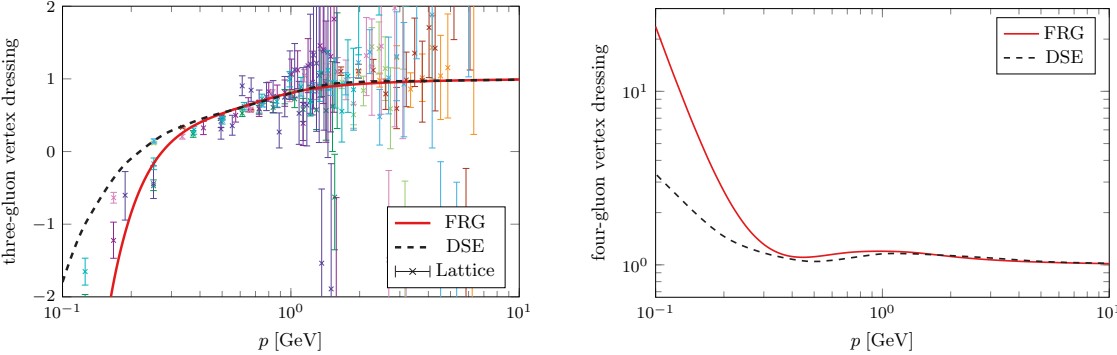

Figure 6: Three-gluon (left) and four-gluon (right) vertex dressings, $\lambda_{A^3}(\bar{p})$ and $\lambda_{A^4}(\bar{p})$, compared to DSE [31] and lattice [11,13,23,51] results.

a larger bump [5]. In order to perform a more informative comparison between the DSE and FRG results, a DSE scaling solution would be preferable because of its uniqueness [58,60].

### 3.2.2  Vertices

The ghost-gluon and gluonic vertex dressings are shown in comparison with DSE [31] and lattice [11,13,23,51] results in Fig. 5 and Fig. 6, where the momentum scale was set using the fit parameters from the gluon propagator in the last section. Similar to the propagators, all dressings converge to unity in the ultraviolet.

Concerning the ghost-gluon vertex dressing, we find that the lattice result has its peak at a higher scale than the dressings computed with functional methods. A similar but, at least in the FRG result less obvious, deviation can be observed already in the ghost propagator dressing, indicating a general scale mismatch between ghost- and glue sector. This is particularly interesting, since also recent QCD investigations with very sophisticated truncation schemes [2,6] show such a scale mismatch between the matter sector and the glue part of the theory, whereas the glue sector in itself runs consistently. We think that in both cases, missing higher-order effects are the most likely source of these deviations.

The FRG three-gluon vertex dressing shows very good agreement with the lattice results over all momenta. In particular, the agreement in the infrared is surprising, since the lattice features a decoupling solution, which has a linearly divergent three-gluon vertex dressing function [29,31,34], whereas our solution is the scaling solution, which has a stronger divergence in the infrared, $\lambda_{A^3}(p) \propto (p^2)^{-3\kappa-1/2}$, cf. subsection 3.3. The FRG and DSE four-gluon vertices agree well, whereas lattice measurements of the four-gluon vertex are not available as of now.

### 3.3  Infrared Scaling Exponents

In the scaling solution, all correlators scale with a specific power law in the infrared. It can be shown that self-consistency demands that the anomalous scaling behavior of any $(2n+m)$-point function with $2n$ ghost and $m$ gluon legs in $d$ dimensions is determined by one single scaling exponent and can be written as [25,58,60]

$$\lim_{p\to 0} \lambda^{(2n,m)}(p) \propto (p^2)^{(n-m)\kappa+(1-n)\left(\frac{d}{2}-2\right)} . \tag{13}$$

In particular, for the two-point functions, the scaling power laws are then given by [53, 54]

$$\Gamma_{\bar{c}c}(p) \propto p^2 \cdot \left(p^2\right)^{\kappa},$$

$$\Gamma_{AA}(p) \propto p^2 \cdot \left(p^2\right)^{-2\kappa + \frac{d}{2} - 2},$$
(14)

where we took their canonical scaling into account. The right panel of Fig. 4 and the left panel of Fig. 5 clearly reveal the power law behaviour. Fitting the propagators with (14), we obtain the three-dimensional scaling exponents,

$$\kappa_{\text{sym. p.}} = 0.321 \pm 0.001,$$

$$\kappa_{\text{full mom.}} = 0.348 \pm 0.013,$$

$$\kappa_{\text{sym. p. + tad.}} = 0.349 \pm 0.003,$$
(15)

for the different truncations. The uncertainty stems from the difference of the ghost and gluon propagator fits. In contrast to the large- and mid-momentum behaviour of the correlators, the scaling coefficient is also susceptible to the full momentum dependence of the vertices.

We also compare these scaling coefficients with those of four-dimensional Yang-Mills theory at finite temperature [7]. There an approximation similar to the symmetric point approximation, (1) in subsection 3.1, was used. Fitting the magnetic part of gluon propagators to the scaling formula (14) yields $\kappa_T = 0.323(3)$. Hence, the magnetic scaling exponent agrees very well with the scaling exponent of the three-dimensional theory in the approximation (1). This is expected from dimensional reduction, and yields a very consistent picture.

## 4   Conclusion

We have presented non-perturbative correlators of three-dimensional Landau-gauge Yang-Mills theory obtained from first principles with the functional renormalisation group. We have checked the reliability of the results by comparing to lattice results and achieved better agreement by including non-classical tensors structures in the truncation scheme. However, at lower momenta the functional and the lattice results still show a discrepancy of 10 %. This hints at sizeable truncation artifacts in three dimensional Yang-Mills theory with functional methods at the current truncation level.

These findings are particularly interesting, because an analogous investigation with the FRG in four dimensions shows considerably better agreement with the corresponding lattice results already at a simpler truncation level, based on classical tensor structures only. This indicates that apparent convergence is achieved with less effort in the four-dimensional theory. A possible explanation are the stronger infrared effects that are generically present in lower dimensions. Phrased differently, the three-dimensional theory features a weakened RG irrelevance of the operators corresponding to the non-classical vertex components.

Interestingly, the effects of non-classical tensors seem to cancel largely. Although individual contributions result in large corrections, their overall effect is relatively small but notable. In this work this is explicitly shown in the propagator tadpole contributions, whose overall effect is small, when compared to the individual contributions. A similar observation has also been made in the matter sector of four-dimensional QCD for the effect of non-classical quark-gluon interactions [2, 6]. This finding is particularly important for devising quickly converging truncation schemes by preserving the underlying resummation patterns.

**Acknowledgments**

We thank Markus Q. Huber and Axel Maas for discussions. This work is supported by ExtreMe Matter Institute (EMMI), the Austrian Science Fund (FWF) through Erwin-Schrödinger-Stipendium No. J3507-N27, the Studienstiftung des deutschen Volkes, the German Research Foundation (DFG) through grants STR 1462/1-1 and MI 2240/1-1, the U.S. Department of Energy under contract de-sc0012704, and in part by the Office of Nuclear Physics in the US Department of Energy's Office of Science under Contract No. DE-AC02-05CH11231. It is part of and supported by the DFG Collaborative Research Centre "SFB 1225 (ISOQUANT)".

# A    Regulator independence

To check the stability of our results, we repeat the computations above with the flat [63] instead of the exponential regulator shape function. We parametrise the ghost and gluon regulators by

$$R^{ab}(p) = p^2 \, \delta^{ab} \, r\left(\frac{p^2}{k^2}\right),$$

$$R_{\mu\nu}^{ab}(p) = p^2 \, \delta^{ab} \, \Pi_{\mu\nu}^{\perp} \, r\left(\frac{p^2}{k^2}\right). \tag{16}$$

The exponential shape function is given by

$$r_{\text{exp}}(x) = \frac{x^{m-1}}{\exp(x^m) - 1}, \tag{17}$$

whereas the flat one is given by

$$r_{\text{flat}}(x) = \left(x^{-1} - 1\right) \cdot \theta\left(x^{-1} - 1\right). \tag{18}$$

The dependence of propagator dressings on the regulator shape functions is shown in Fig. 7 as relative errors, defined by

$$\Delta_{\text{rel}}^2 = 2 \cdot \frac{(\mathscr{O}_{\text{exp}} - \mathscr{O}_{\text{flat}})^2}{\mathscr{O}_{\text{exp}}^2 + \mathscr{O}_{\text{flat}}^2}. \tag{19}$$

Clearly, the relative errors are well below the percent level in the IR, and even smaller in the mid-momentum and UV regimes that are relevant for hadronic observables. Importantly, the regulator dependence is significantly smaller than the truncation dependence.

Explicitly demonstrating regulator independence is a standard quality and self-consistency check for truncations in the FRG. It is a necessary but not sufficient criterion for the convergence of a given truncation. Indeed, we observe that the dependence of our results on the regulator shape function is negligible although the truncations are not yet converged. Nonetheless, this regulator independence already at low truncation orders is a very welcome property.

# B    Numerical computation

Landau gauge has the convenient property that the transverse correlation functions close among themselves [5, 64], i.e. correlators with at least one longitudinal leg do not couple

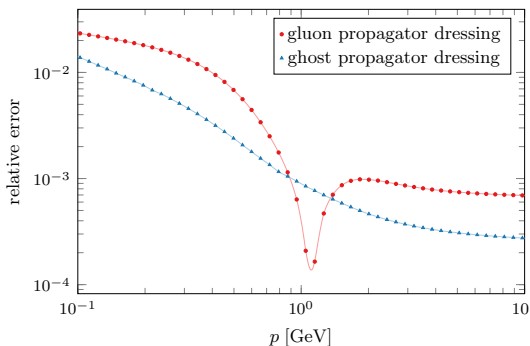

Figure 7: Relative errors $\Delta_{\mathrm{rel}}$ of propagator dressings obtained with different regulator shape functions, given in (17) and (18), in the symmetric point approximation.

back into the transverse subsystem. In the presence of a regulator term, the BRST symmetry is encoded in modified Slavnov-Taylor identities. Their most important consequence is a non-vanishing gluon mass term at finite cutoff scales [65]. Here we present only results for one choice of the gluon mass term, determined uniquely by the scaling solution [53, 54]. The consequences of other choices for the gluon mass term are qualitatively similar to YM theory in four space-time dimensions and we refer to the discussion presented in [5] for details.

This work relies on the workflow established within the fQCD collaboration [1], see [5] for details. Symbolic flow equations were derived using *DoFun* [66], traced using *FormTracer* [67], which makes use of FORM [68] and its optimization procedure [69].

# C  Scale setting and normalisation

For comparison, the DSE and lattice results for the propagators in Sec. 3 are normalised in amplitude and momentum scale relative to the FRG results. To that end we normalise the DSE/lattice gluon dressings with a least squares fit to the FRG gluon propagator dressing in the range 3 GeV to 6 GeV with

$$\min_{c_A, c_p}\Big\{ \sum_{p_{i,\mathrm{lattice}}} \big(c_A\, Z^{-1}_{A,\mathrm{FRG}}(c_p\, p_i) - Z^{-1}_{A,\mathrm{lat/DSE}}(p_i)\big)^2 \Big\}. \tag{20}$$

Here, $c_A$ normalises the amplitude while $c_p$ normalises the momentum scale. The momentum scale normalisation has to be used for all correlation functions. Hence it is only left to fix the amplitudes for the other correlation functions. In particular the amplitude of the ghost propagator dressing is normalised with

$$\min_{c_c}\Big\{ \sum_{p_{i,\mathrm{lattice}}} \big(c_c\, Z^{-1}_{c,\mathrm{FRG}}(c_p\, p_i) - Z^{-1}_{c,\mathrm{lat/DSE}}(p_i)\big)^2 \Big\}. \tag{21}$$

The lattice results for the vertices have large statistical lattice error, and we refrain from normalising the amplitudes. The dressing of the DSE vertices is trivial for large momenta.

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
