# Peer review of "Correlation functions of three-dimensional Yang-Mills theory from the FRG"

_SciPost Physics, doi:SciPost Phys. 5, 066 (2018)_

## Round 2 · Referee Report · Anonymous (Referee 1) · 2018-10-26

Report

The present work contains a detailed study of the complicated dynamics associated with the fundamental Yang-Mills correlation functions (propagators and vertices) in $d=3$. This problem is particularly interesting, given that in $d=3$ the gauge coupling is dimensionful and the theory superrenormalizable, and is believed to be the host of a plethora of highly non-trivial effects. Moreover, in the last decade or so, a multitude of lattice simulations have furnished solid results, against which the underlying approximations and theoretical assumptions may be tested.

The basic theoretical framework employed for this particular analysis is the "functional renormalization group", which has been established over the years as one of the main non-perturbative approaches in the continuum. The authors are world-class experts in this particular field, and the present work is therefore of the highest quality.

From the technical point of view, the results of their analysis are in general quite compatible with (without being a subset of) similar studies carried out in the context of Schwinger-Dyson equations; these works have been most appropriately cited by the authors. To be sure, the authors find minor differences, as they point out in considerable detail, which may be expected, given the wide array of truncation schemes employed in the cited literature. In the case of the vertices, the authors attribute the observed "deviations" to "missing higher order effects", which is, of course, a most plausible explanation. In addition, they almost exclusively focus on the "gapped scaling solutions", which might also partially account for some of the observed differences.

In my opinion this manuscript is well written and contains a particularly useful contribution to the ongoing search into the infrared dynamics of Yang-Mills theories. I therefore recommend its publication in its present form.

---

## Editorial Decision

published